# The Cosmopolitanism of the Early Sophists: The Case of Hippias and Antiphon

Giovanni Giorgini 

Department of Political and Social Sciences, University of Bologna, I-40125 Bologna, Italy;
giovanni.giorgini@unibo.it

**Abstract:** An investigation of the emergence of the notion of 'Cosmopolitanism' in 5th century Greece. The author focusses on the early sophists, and specifically on Antiphon and Hippias.

**Keywords:** cosmopolitanism; sophists; ancient political thought

## 1. Introduction

What does it mean to be a *kosmopolites*, a "citizen of the world"? The first occurrence of the word in Greek is attributed to the Cynic philosopher, Diogenes of Sinope by Diogenes Laertius, who reports that when "asked where he came from, he said 'I am a citizen of the world".[1] Diogenes of Sinope was famous for his quips and for his outrageous and provocative answers. It is possible that he wanted to shock his interlocutors by using a word that sounded like an oxymoron: being a citizen meant belonging to a city, a specific political community. The expression typically used for this status was *metechein tes poleos*, "participating in the city". How is it possible to belong to, or participate in,[2] the world? While the phrase "citizen of the world" does not appear in the available sources concerning the sophists, in this paper I will argue that at least two sophists anticipate Diogenes' stance, namely Hippias and Antiphon. After briefly describing what citizenship meant in ancient Greece and elucidating the meaning of "cosmopolitanism", this essay will examine the literary sources concerning two famous 5th century BCE sophists, Hippias and Antiphon, to see whether there is evidence of their proposing a cosmopolitan view. I will also address the question of what kind of cosmopolitanism they were allegedly proposing—the view of the unity of mankind or, alternatively, the brotherhood of wise men. I will conclude with an evaluation of the innovativeness of their theories in the context of 5th-century Greek political culture.

Originally, the word *politai*, "citizens", referred to "the inhabitants of the city" and was thus connected to a geographical place. However, being a citizen in the time of Diogenes of Sinope (the 4th century BCE) not only meant living in a city and enjoying full civic and political rights (and obligations), it also meant participating in public activities, including serving in the army and celebrating religious festivals. Walter Burkert has maintained that citizenship meant, first and foremost, *koinonia ton hieron*, "community of sacred matters". Conversely, *atimia* meant exclusion from sacred matters.[3] Being a citizen was thus a complex concept that included personal identity, legal and political conditions, and recognition of a specific status (*timé*). Hence, the act of being stripped of one's citizenship (*atimia*) not only meant being disenfranchised but also being deprived of one's status, identity, and ability to interact with others in one's community. Being a citizen was also a privilege, as indicated by the practice of Greek cities to grant citizenship to foreigners only for exceptional merits. Conversely, it was a typical tyrannical measure to enfranchise foreign soldiers, merchants, and other useful people to reinforce and stabilize their hold on power. This policy generated, in certain turbulent periods, a process of revision of the citizens' list and the disenfranchisement of "spurious" citizens. Perhaps the most famous such measure

was the *diapsephismos* Isagoras, proposed after the Pisistratid tyrants were chased from Athens in 511 BCE (see Cartledge 2016; Giorgini 2019).

I believe we should interpret Thucydides' statement that "the men [are] the city" (*The Peloponnesian Wars* 7.77.7) literally rather than metaphorically: up until the middle of the 4th century BCE, the words for "citizen" in Greek—*polites* and *astos*—appeared mostly in their plural form because citizens were conceived as a plurality. The identification of the citizen with the city is also reflected in the common practice of referring to a political entity by the name of the citizens, not the city: the war between Sparta and Athens, for example, is described in Thucydides' famous opening lines as "the war fought by the Peloponnesians and the Athenians" (1.1). There was thus a strong sense of identity with one's city and, in the case of Athens after Cleisthenes's reforms, even with one's *deme*, which was the smaller administrative unit specially created to foster a sense of belonging in citizens living in different parts of Attica. In addition, there were traditional as well as legal norms that regulated the status of resident aliens (metics) and foreigners (*xenoi*): *xenia* was a kind of institutionalised hospitality that took the form of ritualised friendship, which served the purpose of connecting eminent families in different cities as well as enabling informal relationships between cities, even during wartime.

In such a localistic, strongly identarian culture, how was it possible to call oneself, and be, a *kosmopolites*, a "citizen of the world"? It all started with the innovative teaching of the sophists. When they made their entrance onto the intellectual and political scene in Athens (and in other important cities in mainland Greece and Sicily) around the middle of the 5th century BCE, they presented themselves as teachers of the art of rhetoric. The great English historian George Grote commented that they filled a vacuum in "higher education" in the city; they also answered the need for experts capable of teaching affluent citizens how to be effective with speech in politics and in court. From its inception, the two main ideological pillars of Athenian democracy were "government through speech" and "equal possibility to speak" (*isegoria*), as well as the "possibility to speak one's mind" (*parrhesia*). These two maxims were almost synonymous with democracy. Other factors made the ability to speak publicly an invaluable skill. For example, the legal institutions that characterised democracy made society very conflictual and antagonistic. Lawsuits and court hearings were commonplace in the life of the ordinary Athenian citizen, so the ability to speak persuasively and argue effectively was crucial. Grote also noticed that after the Ionic revolt (500 BCE) and the Persian invasions of Greece (490–479), the relations between Greek cities became more frequent and more complicated, requiring more talent—especially rhetorical skills—from the politicians who managed them.[4] Training in persuasive speech was therefore as equally essential as training in arms for a Greek citizen, and the sophists claimed to be able to provide exactly such an education.

The sophists were not a school or a movement, strictly speaking, but they had some research interests in common and a general "rationalistic" approach: they all believed, in a rather aggressive way that anticipates (for some scholars) the iconoclastic attitude of the seventeenth century Enlightenment philosophers, that the unfettered use of reason was the key to arriving at the truth of matters, whatever that was.[5] In addition, the sophists were proud to proclaim their conclusions regardless of how shocking and subversive they were in comparison to traditional beliefs and values: received opinions were regarded as prejudices, and nothing could stand unchallenged before the bright light of reason.[6] They created, and interpreted well, the spirit of the age, which was a critical spirit in its etymological meaning: *krisis*, the rational evaluation of reality in all its facets. It is no wonder then that their works often bore the title *On Truth*: they were proud to proclaim that their rational inquiry had led them to discover the truth about some matter, although they challenged the correspondence view of truth demonstrated by Protagoras' relativism and Gorgias' claim that being cannot be communicated. This fact inevitably put them in conflict with the traditional "masters of truth" of Greece: the poets, whose teaching the sophists challenged and eventually replaced,[7] and the diviners and soothsayers, whose methods they rendered obsolete and ridiculed. Their opinions and practices often upset

ordinary citizens, who associated them with the wealthy and with criticism of the shared values of the *polis*. The sophists charged a fee for their teaching, which meant that only wealthy people could afford to pay for their services. They also employed their verbal skills to examine and question the traditional beliefs of their contemporaries. This uneasiness, if not outright condemnation, explains, for instance, the 432 BCE decree named after the soothsayer Diopeithes, which punished people who scrutinised *ta meteora*, the celestial bodies (see Plutarch, *Pericles* 32.1; Dover 1976). It is also reflected in Aristophanes' bitterly mocking portrait of Socrates in *Clouds*, which depicts him as a "sophist" at home in the clouds, imparting esoteric and dangerous counsels to the acolytes in his school or Thinkery (*phrontisterion*) (see Notomi in this Special Issue).[8]

Among these innovative ideas was the momentous distinction between nature (*physis*) and law (*nomos*). Historically, this distinction was most likely the result of a generalization of some observations made by Greek sailors and merchants, who observed that laws and customs in other countries were different from, and sometimes opposite to, those of the Greeks. Knowledge of other people's different religious beliefs, legal systems, and political arrangements questioned the previously firm belief that human society and its arrangement were a reflection of the order of the universe (*kosmos*). Law became synonymous with what is particular, located in time and place, and nature with what is universal and valid everywhere. As a result, God and the justice of Zeus ceased to be viewed as the foundation of the order in the world, and nature (*physis*) became the criterion of general validity and universality. This realization prompted two opposite reactions. Some authors emphasised the "cultural" side of the distinction and therefore the differences: they maintained that "custom [*nomos*] is the king of all things" (Herodotus, *Histories* 3.38) and that "Of all things the measure is man: of those that are, that they are; and of those that are not, that they are not" (Protagoras DK 80 B1). Laws, institutions, and traditions are inevitably linked to society, for there is an evident contrast between what is valid by nature, always and everywhere, and what is valid by custom or law and is therefore situated in a specific time and place. Relativism was the logical conclusion of these premises: morality and law are "the opinion of the city", as Protagoras puts it in Plato's *Theaetetus* (167c). Conversely, some authors emphasised the "natural" side of the distinction and therefore the similarities. Beyond the cultural differences, they argued, there lies something common to all human beings—human nature. Human nature unites us all across cultural and political barriers: universalism, cosmopolitanism, and the emergence of the notion of "natural" law are the results of this alternative line of reasoning. It is this second intellectual trend that I intend to investigate in this essay, focusing on the work of two sophists from the fifth century BCE: Hippias and Antiphon.

## 2. Hippias, Sophist, Ambassador and Traveller

Hippias of Elis (ca. 460–ca. 390 BCE) travelled extensively throughout Greece as an ambassador for his native city, as well as to give public speeches and participate in Panhellenic events, in which he displayed his vast learning and rhetorical ability.[9] He came to Athens for the first time probably around 430. He wrote on many different subjects, from astronomy and music to linguistics and painting, and was famous for his ability in the art of memory. Among his publications was a collection of excerpts from ancient philosophers and poets, both Greek and non-Greek—perhaps a hint to his universalist view of mankind and ecumenical notion of culture. This supposition is based on Clement of Alexandria's *Stromata*, in which Hippias declares, "Of these [*scil*. probably: ancient opinions] some have doubtless been expressed by Orpheus, others by Musaeus, to put it briefly, by each one in a different place, others by Hesiod, others by Homer, others by the other poets; others in treatises; some by Greeks, others by barbarians. But I myself have put together from out of all these the ones that are most important and are akin to one another, and on their basis, I shall compose the following new and variegated discourse" (6.15; trans. Laks-Most 2016; slightly altered).

It is hard to build a case on such scant evidence, but Hippias' reference to his equal use of Greek and barbarian contributions, deemed to be on the same qualitative level, seems to point to a cosmopolitan view of culture. This is unusual, to say the least, in the second part of the 5th century BCE, when the cultural operation that had taken place after the Persian wars—the creation of the character of "the barbarian" as the inferior Other—was still underway.[10] Hippias maintains that he can compose a "new and variegated discourse"[11] building on Greek and barbarian sources: evidently the "barbarians" could contribute to world civilization, as they always did before the ideological opposition brought about by the Persian wars determined "the closing of the sluice" or end of Asian influence on Greek culture.[12] To corroborate this hypothesis, we can adduce two other pieces of evidence. First, Hippias is described by Plato and other sources as a very busy diplomat and lecturer who travelled the world extensively: he had first-hand knowledge of many different alien cultures and was able to appreciate "alien wisdom".[13] He was a successful "man of the world" (perhaps even a *kosmopolites*) who felt at home everywhere and perhaps even spoke a foreign language. The second piece of evidence comes from Plato's *Hippias Minor* and is thus slightly clouded by Socrates's famous irony. Here Socrates recounts hearing Hippias boasting of his wisdom and expertise (*sophia*) "in the marketplace near the bankers' counters". More specifically, Hippias boasts that he once went to Olympia (presumably for the general meeting of the Greeks) wearing only things that he had made himself: "First, the ring that you were wearing (for that is what you began with) you had made yourself, as you knew how to engrave rings [ . . . ] And what seemed to be the most extraordinary thing to everyone, and the demonstration of the greatest expertise [*sophia*]: you said that the girdle of the tunic you were wearing was like the very luxurious Persian ones, but that you had plaited it yourself" (*Hippias Minor* 368b–c). Socrates is flattering Hippias while driving him into a dialogical cul-de-sac: he will find himself arguing that the person who voluntarily errs and does disgraceful and unjust acts is the good man (376b). What is remarkable in this passage is Socrates' allusion to the fact that Hippias is dressed in attire fashionable in Persia and practicing the typically Asian habit for men of wearing rings. Socrates underlines that Hippias began his egotistical narration with the ring in order to point out Hippias' penchant for indulging in non-Greek customs. In addition, in contrast to the restrained behaviour of the Greeks, Hippias sported a luxurious Persian-style girdle: he obviously knew and appreciated this kind of garment and was even capable of manufacturing one himself.[14]

More generally, in *Hippias Major* (284b) and *Hippias Minor* (363c–d), Hippias is introduced as a self-confident and conceited person who boasts of his ability to engage in both political and intellectual endeavours and carry on public and private affairs. For example, when asked by Socrates about his prowess in rhetoric, he says that he delivered excellent epideictic displays of his ability in public discourses at Sparta but was not allowed to educate their youth because their laws do not permit education by foreigners (284c). Here, he also boasts of his wealth: "You may be sure that if anybody had ever received money there in payment for education, I should have received by far the most". Socrates then asks Hippias whether he considers the law an injury to the city or a benefit, to which Hippias replies that it is a benefit, but then fails to realize that if the law is not based on the true notion of justice, then nothing is certain, everything is debatable, there is no truth in politics or in court, and people have to resort to persuasion.

Hippias also appears as an interlocutor in Plato's *Protagoras*. He is among the many sophists gathered into the house of the rich Callias and speaks at a peak moment of the dialogue: the discussion between Socrates and Protagoras has come to a halt because Protagoras has given a long speech and Socrates prefers the short question-and-answer format and therefore threatens to leave. Their skirmishes are paused by the host's intervention, who entreats Socrates to stay. Some of the people present make suggestions about how to proceed, including Alcibiades, Critias, and Prodicus. At this point, Hippias makes a conciliatory proposal and suggests finding someone to act as an umpire. He is introduced by Plato as "Hippias the wise [*sophos*]" and he proceeds to deliver this discourse:

Gentlemen who are present here, I consider that you all belong to the same family [*syggeneis*], household [*oikeious*], and city [*politas*]—by nature [*physis*], not by convention [*nomos*]: for what is similar belongs by nature to the same family [*syggenes*] as what is similar, whereas convention, which is a tyrant over men, commits violence upon many things against nature. Therefore, it would be disgraceful for us to know the nature of things [*physin ton pragmaton*]—we who are the wisest of the Greeks and have come together now to [*scil.* the city] that is, in Greece, the town hall itself of wisdom and, in that city itself, to this house, the greatest and most wealthy one it contains—but not to produce anything that would be worthy of this honour, but instead to quarrel with one another like the most vulgar of men (*Protagoras* 337c–338).

Here, Hippias plays on the typically sophistic opposition between nature (*physis*) and convention (*nomos*) to argue for the similarity of all human beings. In a progression, he states that human beings are similar (*homoioi*) and therefore belong by nature to the same family, household, and city. Convention, on the contrary, shatters this natural unity and creates artificial, forced differences: Greek and barbarian (we may surmise), citizen and slave, nobleman and commoner, and so on. Hippias' statement is revolutionary and potentially subversive. It is also remarkable that he compares convention to a tyrant, since it was commonplace in Athenian culture from Cleisthenes onwards to describe the law, *nomos*, as the bulwark and safeguard against tyranny: the objective law stands firm opposite the arbitrary will of the tyrant. Hippias evidently does this to draw the attention of the listeners and to dramatize this opposition between *physis* and *nomos*, for all his contemporaries would have been struck by this comparison. This conventionalist view of law is reiterated in Xenophon's *Memorabilia*, in which Hippias argues that "the laws of the city" are "what the citizens have written down after having made an agreement about what people must do and what they must refrain from". He then concludes: "Socrates, how could someone consider either the laws or obedience to them to be something worth taking seriously, given that the very people who establish them often reject and change them?" (4.4.7–14).

I described Hippias' statement as revolutionary and potentially subversive but I should add a few words of caution. How unprecedented was Hippias's statement? Was it universal in scope or restricted to wise men? Furthermore, how serious was it? Although Hippias' view of the unity of mankind is daring, especially in the ideological conditions of the time, it is not unique or unparalleled. In the second half of the 5th century, Hippocratic medicine had already made the momentous discovery that human bodies react in the same way to the same substances: Greek and barbarians, men and women, freemen and slaves are all weakened and eventually killed by abstinence from food (*De antiqua medicina* 9). The author of *De antiqua medicina*, one of the earliest medical treatises, speaks of "the nature and power of man" (*he tou anthropou physis te kai dynamis* [3]), which is the foundation for finding the appropriate diet and treatment for a patient. Certainly, there exist some differences, so that sick and healthy people have different requirements and certain foods are not good for some people, but there are some general rules based on the existence of a shared human nature, and ailments present themselves with the same symptoms everywhere.[15] The author even gives an example of a general rule: "Undiluted wine, drunk in large quantity, produces a certain effect upon a human being" (*De antiqua medicina* 20). It is upon this universal basis that the good physician uses his own judgement and perception to evaluate the specific case at hand. The results of Hippocratic medicine, which became the most advanced science of the age (as the author of *De antiqua medicina* proudly asserts), were so impressive that Thucydides built his science of history upon their method and results. His narration of the war between the Spartans and the Athenians is constructed on the premise that there exists a universal human nature (*anthropeia physis* [3.45.7]; similar at 3.82.2) or "human condition" (*to anthropinon* [1.22.4]) characterised by the desire to have more and more (*pleonexia*), which prompts men to act according to certain patterns. This is even described as a "necessary nature" (*physis anankaia* [5.105.2]) in the dramatic exchange between the Athenian generals and the Melian oligarchs. The effect of this universal desire

to aggrandize is that wars are inevitable when the balance of power is tilted in one direction. Thucydides can accordingly state his opinion that the truest cause (*alethestate prophasis*) of the Peloponnesian war was the increase in Athenian power, which "forced" (*anankasai*) the frightened Spartans to declare war (1.23.6).

Hippias seems to go one step further in finding a universal similarity among human beings who belong to the same family, household, and *polis*. However, does this statement apply to all mankind? I believe his opening words in *Protagoras* prove that Hippias was referring only to the intellectual elite present in Callias's house. He addresses his audience as "Gentlemen who are present here" (337c), and we may infer that he had in mind a philosophical *polis* to which all sophists and wise people belong: Greek and non-Greek, to be sure, but still a restricted, eminent, intellectual group. This restrictive interpretation better suits the notion that "all present people" participate in the same cosmopolitan city. It also fits better with the testimonies in Plato's *Hippias Minor* and Clement of Alexandria's *Stromata* discussed previously: both passages point to the equal value of Greek and non-Greek intellectual, or practical, products. This is the ideal of the wise man (*sophos*), who knows the nature of the world and is therefore united with all wise men, that we will find clearly illustrated in Stoic philosophy.[16]

### 3. Antiphon, Rhetorician and Sophist

It was another sophist, Antiphon (ca. 480–411), who took the next step towards cosmopolitanism and the view of the unity of mankind. Antiphon had a many-sided personality and displayed his talent in many fields. He was a native Athenian and a slightly older contemporary of Socrates. According to Xenophon, the two were rival teachers, but this is doubtful in view of the generally apologetic tone of the work.[17] Sadly, Antiphon and Socrates were somewhat associated in death because, a few years apart, they were executed under the newly restored Athenian democracy. Indeed, both used their trial to deliver a defence that was in fact an apology of their entire way of life, but both failed to persuade the jurors and decided to accept their fate instead of fleeing the city.

The diversity of the many writings attributed to Antiphon is a testament to his curiosity and ability to write on a variety of subjects such as politics and the interpretation of dreams, medicine, trials for personal or political matters, and the nature of discourse itself. He is even credited with having developed an early form of psychoanalysis—an "art of eliminating pain" through discourses (DK 87 A6). In this, he evidently shared Gorgias's view of the power of speech (*logos*) as a *pharmakon*, which works as an incantation and creates in the soul the same effects that drugs create in the body (DK 82 B11). Since antiquity, this fact, together with the differences in style of the surviving work, has led interpreters to wonder whether one single person could be the author. Already in the first century BCE, for example, the grammarian Didymus spoke of two Antiphons: the logographer/orator and the sophist (whose biography is, however, completely unknown). Many other interpreters accepted this position and, thanks to the diffusion of the name, sometimes even added a third (or more) Antiphon.[18] In recent decades, however, the view that the works attributed to Antiphon have only one author has gained wider acceptance. The sensible and cogent arguments put forth by Michael Gagarin seem to me conclusive in supporting a unitarian view of Antiphon's identity.[19]

Antiphon was the first person to write speeches for other people to use in court (he was a *logographos*), while he himself did not like to speak in public (Thucydides, *The Peloponnesian Wars* 8.68). According to Plutarch (*Lives of the Ten Orators* 10 = DK 87 A3), he served Athens in many roles, including as a trierarch of two ships, a general, and an ambassador, and although he had played an active role in Athenian political life, his defining moment came when he was involved in the oligarchic coup of 411 BCE. It is interesting that Antiphon chose a moment that could have become an historical turning point in Athenian history to act. In retrospect, the events of 411 look like a failed coup that was quickly suppressed by the democrats, but at the time, the coup roused the interest of many prominent citizens who were disgruntled with the democratic government and tired

of the long war against Sparta. Thucydides' opinion on the events leaves no doubt: the attempt to change the regime in Athens was plotted and carried out by some of the finest men in the city, such as Theramenes ("a man of considerable eloquence and intellectual power"), Peisander, and Antiphon himself, described as "a man second to none of the Athenians of his time in excellence and one who was extremely forceful in thinking and in saying what he thought" (*The Peloponnesian Wars* 8.68). The enterprise was inherently difficult: "It was not an easy task to terminate the liberty of the Athenian people almost exactly a hundred years after the deposition of the tyrants, when they had been not only free of subjection to anyone else but also, for over half of that period, accustomed to imperial power over others". (8.68)[20]

Thucydides also provides another interesting piece of information about Antiphon (*The Peloponnesian Wars* 8.68): like many sophists, he was regarded with suspicion by many for his reputation for cleverness (*deinotates*). This judgement may reflect Thucydides' bias against the Athenian populace and its anti-intellectualism, but it is surely true that in his defence at his trial, Antiphon tackled the accusation that he used his wits and persuasive skills like a sophist and wrote discourses for others at a great profit. Indeed, he turned the tables on his accusers and maintained that he could profit from his talent only in a democracy and therefore had no incentive to overthrow the regime. In fact, the events of 411 can be construed as an attempt to reverse the drift towards radicalism of Athenian democracy and to restore a milder version that would allow aristocrats to engage in public life again. This ideal is captured by the expression *patrios politeia*, which vaguely refers to the previous democratic regimes presided over by Solon and Cleisthenes.[21] Thucydides' judgment is, again, very clear, "And now for the first time, at least in my lifetime, the Athenians enjoyed a political system of substantial and obvious merit, which blended the interests of the few and the many without extremes, and began to restore the city from the wretched situation into which it had fallen" (*The Peloponnesian Wars* 8.97). Antiphon thus appears to have been a moderate oligarch, or even a moderate democrat, who eschewed extremes and loved his country more than partisan politics, which was not the rule in those dire days.[22]

In Antiphon we are thus dealing with a complex and multi-faceted personality. Antiphon's political views seem to be a consequence of his wide-ranging reflections on the nature of the universe, especially on the relationship between nature and culture. This seems to be the general theme of his famous work, *On Truth*, which duly forces listeners and readers to question the veracity of the common beliefs and practices of the Greeks. One of the longest surviving fragments of this work begins with an apparently traditional account of justice: "Justice consists in not transgressing the legal institutions [*nomima*] of whatever city one happens to be a citizen of". From this premise, however, Antiphon draws a very subversive consequence:

Therefore, a man would make use of justice in the way that would be most advantageous for himself if, in the presence of witnesses, he considered that it is the laws [*nomoi*] that are great, but, alone and without witnesses, it is what belongs to nature [*physis*]. For what belongs to the laws is <adventi>tious, but what belongs to nature is necessary. Furthermore, what belongs to the laws is the product of an agreement, not of nature, but what <belongs to nature> is the product of nature, not of an agreement [Col. 2]. Therefore, if someone transgresses against legal institutions without being noticed by those who agreed upon them, he escapes shame and punishment. However, if they notice, he does not. In contrast, if, contravening what is possible, he does violence to anything produced by nature, the harm is not less if no man notices him, and it is not greater if all men see him, for it is not opinion that harms him, but the truth. Our examination of all these points brings us to the following thesis: things that are established simply according to the law are established in a way that is hostile to nature.

DK 87 B44 = POxy XI, 1364A; trans. Laks-Most 2016 (slightly altered)

Antiphon goes on to enumerate examples of the opposition between law (*nomos*) and nature (*physis*), arguing that "what is established by the laws are fetters upon nature, while what is established by nature is free". He argues that by transgressing the demands of nature, a person suffers real damage, one that "is not in appearance but in truth". Antiphon's conclusion that many of the things that are just according to the law are at variance with nature shows that he was in fact thinking of a double opposition: truth and opinion are different and opposed, much like nature and law. Consequently, by looking at what happens in nature, one can apprehend the truth about the world, including the human realm.[23]

Antiphon can thus attack another staple of Greek culture in the fifth and fourth centuries BCE—the opposition between Greeks and barbarians. He does so by appealing to the observation of nature and using quite matter-of-fact arguments: "In this behaviour we have become like barbarians towards each other, when in fact by nature we all have the same nature in all particulars, barbarians and Greeks. We have only to consider the things which are natural and necessary to all mankind. These are open to all [to get] in the same way, and in [all] these there is no distinction of barbarian or Greek. For we all breathe out into the air by the mouth and the nose, and we all work with our hands and we walk with our feet" (DK 87 B 44 = POxy XI, 1364B). Here, Antiphon questions the opposition between Greeks and barbarians by appealing to the evident fact that all human beings share the same body. He argues for the existence of an underlying, single human nature, characterised by the same features and exigencies present in all human beings. He also plays on the word "barbarian", which originally referred to speaking broken Greek and thus being incomprehensible.[24] Human beings have "become like barbarians towards each other [*bebarbarometha*]" because they no longer understand each other: they dwell upon the differences between them and overlook their profound underlying similarity. For Antiphon, it would have been enough to look at the body that we all share to see the truth.

In the same work, *On Truth*, Antiphon argues that testifying in court against someone, even if telling the truth, causes harm to that person and therefore is against the norm of justice that one should not wrong anyone. Legal justice has an additional problem: he who seeks justice in court must persuade the jurors that he has been wronged, but the opponent has the same opportunity, for "persuasiveness is balanced" between the accuser and the accused. Antiphon thus concludes that "the administration of law and justice and arbitration with a view to a final settlement are all contrary to justice" (DK 87 B 44 = POxy XV, 1797). I believe that Antiphon drew these conclusions from his practice in court, much like Protagoras and Gorgias had done before him, by watching the meetings of the assembly and the workings of trials in Athens. These two contexts, characterised by adversarial arguments, disclose the importance of persuasion as well as the elusive meaning of "truth". On one hand, they show that truth is ineffective if it is not persuasive; on the other, they show that truth is not the correspondence of something with reality. For example, in court the "truth of the matter" does not necessarily reflect the actual truth of what has happened, but is rather the result of the proceeding itself: the "truth" lies in the verdict, which is the result of the debate between the competing arguments of the prosecution and the defence, and persuasion is therefore of the essence. Protagoras derived from this realization the view that is at the foundation of his *Antilogiai*: the notion that there are two opposed arguments about any topic (see Giombini and Reames in this Special Issue). This realization discloses the complex, dramatic nature of reality, the different perspectives one may adopt to observe and judge an event, and reveals the sense of dismay at this loss of innocence about the "reality" that surrounds us. Likewise, Gorgias's famous and preposterous statements in *On Nature or On Not-Being*: "nothing is; and even if something is, it cannot be known; and even if it can be known it cannot be communicated to someone else", can be construed as the result of his observation of Athenian everyday life: truth is the result of an *agon*, a competitive confrontation of rival arguments, be it a trial in court, a debate in the market-

place, or a philosophical argument. This realization leads to the view that there is no true account of the world, and to the conclusion that, if we cannot invoke truth, we must rely on the power of speech to persuade. As Gorgias phrases it in his *Encomium of Helen*, speech is a "powerful lord", and its "effect upon the condition of the soul is comparable to the power of drugs over the nature of bodies" (14 = DK 82 B11). Hence the importance of rhetoric, since truth is not a property of the world, it is up to rhetoric, in an agonistic context, to persuade people of the truth.

As for Antiphon, we can try to draw some conclusions about his criticism of laws and institutions through his support of the validity and force of nature, although with the usual proviso that it is both difficult and dangerous to extract a theory from such fragmentary evidence.[25] We may first observe that he was a legal professional (a logographer or legal "ghost-writer"), and his examples from the practice of law are telling: he developed his general opinions—his theory of justice—in the concrete circumstances of legal practice in Athens, noticing that what the laws of the city ask us to do conflict with the injunctions we may derive from observing nature and from an absolute notion of justice. Now, nature is both a descriptive and a prescriptive notion (since we derive what we ought to do from the observation of how things work in nature), and Antiphon attacks *nomoi* and *nomima* both for their lack of universality and for being disadvantageous to those who abide by them: if we look at the truth in a clear-eyed fashion, "the just" and "the useful" (what is advantageous to us) are often opposed. The only sensible conclusion is that the truth is that all human beings share a common nature and we should therefore follow its intimations. Antiphon's views open the way to cosmopolitanism and to utilitarianism, two disruptive theories in the polis-centred Greek morality of his day.

Theory and practice are intertwined but sometimes at variance in Antiphon. After all, how could a "lawyer" who believed that legal justice was against nature continue to practice law? It is perhaps fair to conclude our examination of his theories by mentioning that he also wrote a work on concord, entitled *Homonoia*,[26] in the heated climate of the year 411, when the oligarchs in power had to negotiate with the democratic hoplites in the Piraeus in order to face the Spartan threat. Since we possess only fragments of this work, I venture to make only two observations. First, it is telling about Antiphon's moderate political attitude that he wrote on such a subject. In it, we read that "there is no worse evil for human beings than *anarchia*":[27] *anarchia* is what happened in Athens at the height of the civil wars following Solon's reforms, and it paved the way to Pisistratus's tyranny.[28] Antiphon thus shows moderation and patriotism in his political views, although we cannot say much more. Second, the surviving fragments give the impression of being part of a very elaborate work, full of existential observations[29] as well as moral and educational recommendations,[30] and not just of a text composed for the occasion. Antiphon's approach to the subject is complex, and it is likely that education and the discoveries of the new medicine played some part in his advocacy of political reconciliation. It is tempting to think that Antiphon shared the view on concord of Xenophon, an author with a similar frame of mind and political ideas. Xenophon wrote that *homonoia* is the greatest asset for a city: it does not consist of sharing the same taste about poetry or theatre but rather of obeying the same laws (*Memorabilia* 4.4.16). But whatever views Antiphon held on the topic of concord, we may be sure that this cosmopolitan thinker, like Hippias and other sophists, gave an original and nonconventional contribution to the topic.

By way of conclusion, we may pose a more fundamental question concerning Hippias, Antiphon, and the sophists more generally: to what extent did they really mean what they stated? Plato, we know, was the boldest thinker, loyal to his argument, which he pursued to its logical conclusion, undeterred by the consequences and unmoved by the common beliefs of his age. In the case of the sophists, we are sometimes left with the impression that they wanted to shock their audience with their innovative and astounding discourses more than state a newly discovered truth. It seems unlikely that, like Socrates, they were prepared to die for their convictions.

**Funding:** This research received no external funding.

**Institutional Review Board Statement:** Not applicable.

**Informed Consent Statement:** Not applicable.

**Conflicts of Interest:** The author declares no conflict of interest.

## Notes

1    D.L. 2.63. Diogenes of Sinope was perhaps preceded by Anaxagoras, at least if we interpret DK A1 as a statement of cosmopolitanism: "When someone asked 'Have you no concern for your fatherland?', 'Be silent,' he replied, 'I am greatly concerned with my fatherland', and pointed to the sky". See (Baldry 1965).

2    See the very detailed analysis of Blok (2005), who is also interested in gender issues about citizenship and warns against Aristotle's restrictive interpretation of the word.

3    Burkert (1995); cf. Andocides 1.71; 32.

4    Grote (1899, p. 338). The first edition was published between 1846 and 1856.

5    I am here thinking of such works as Theodor Gomperz's influential *Griechische Denker* (1896–1909; Engl. trans. London: J. Murray, 1901–1912), which described the impact of philosophy, and the sophists, on 5th century Greek society as "the age of Enlightenment", or the third volume of W. K. C. Guthrie's *History of Greek Philosophy* titled *The Fifth-Century Enlightenment*: Guthrie (1969). See also Dodds (1951).

6    If this sounds too literary and emphatic, one may see the aggressive stance displayed by Eteocles in his confrontation with Polinices in Euripides's *Phoenician Women*: Eteocles speaks with the force of the newfound truth he has reached with his reasoning. Conversely, one may observe the qualms that this "rationalistic" attitude induced in many people in Sophocles' dramatic treatment of Oedipus, who solved the enigma of the Sphinx with his reason unaided by supernatural forces, only to discover his own condition of *paida tes tyches*.

7    See for instance Protagoras's statement that "the greatest part of a man's education is to be clever about poetry" (Plato, *Protagoras* 338e). Plato himself followed in the wake of the sophists in challenging the poets' pretences about truth.

8    To mention only the most famous example. The Attic comedy is replete with lost plays mocking the "new intellectuals", their eager patrons and their gullible pupils.

9    DK 86 A 2.

10   See Hall (1989) and Giorgini (2002). Hall focuses on the representation of the "barbarian" in Greek tragedy, while Giorgini examines the political origin of this immensely successful cultural operation that transformed the neighbour into the inferior Other.

11   Hippias uses *polueide* for "variegated". It is very interesting that in another testimony from Philostratus (*V. Soph.* 1.11 = DK 86 A 2) we read that Hippias "charmed Greece at Olympia with stylistically variegated (*poikilois*) and well-meditated speeches". Another hint to Hippias' use of multifarious sources?

12   I am borrowing M. L. West's expression from his pioneering book, *Early Greek Philosophy and the Orient*: see (West 1971).

13   This is the title of Arnaldo Momigliano's famous book, *Alien Wisdom*: see Momigliano (1976). Hippias's writings included a work titled *The Names of Peoples* (DK 87 B2).

14   In *Varia Historia*, Aelian observes that "the story is widely reported that Hippias and Gorgias appeared in public garbed in purple attire" (12.32 = DK 82 A9).

15   See for instance [Hippocrates], *Prognostic* 25: "It must be clearly realised with regard to symptoms, certain and otherwise, that in every year and every region bad signs have a bad significance and good ones a favourable implication; for the symptoms mentioned above prove valid in Libya, in Delos, and in Scythia".

16   Mario Untersteiner, in his landmark edition of the sophists, argues for a universalistic interpretation of this statement which, in his opinion, discloses a cosmopolitanism beyond the differences created by positive law and based on nature: See Untersteiner (1954, pp. 104–5) with footnotes.

17   Xenophon, *Memorabilia* 1.6. This opinion seems to be corroborated by Diogenes Laertius, who quotes Aristotle on this (D.L. 2.46 = fr. 75 Rose).

18   See also the opinion of Hermogenes (2nd century CE) in DK 87 A2. On this see Narcy (1996).

19   See Gagarin (2002). André Laks and Glenn Most, in their recent edition of the fragments of the early Greek philosophers, attribute the fragments to a single Antiphon (Laks and Most 2016, pp. 2–3). See also Pendrick (1993).

20   See also the allusions to the event in Adeimantus's discourse in Plato, *Republic* II, 365d.

21   Compare the use of this expression in the sophist Thrasymachus. See Cartledge (2016) for the historical context.

22   The epitome of partisan politics is perhaps Thucydides' depiction of factional strife in Corcyra (*Peloponnesian War* 3.80–82), but many other events show that party allegiance was by many deemed more important than any other kind of tie, including family ties.

23      I take *physis* to be a normative concept in Antiphon. For a different opinion one may see the very good treatment in Bonazzi (2021).

24      For this original meaning see Strabo 14. 2. 28. Heraclitus (DK 22 B107) maintained that "bad witnesses for humans are the eyes and ears of those who possess barbarian souls", namely who do not understand the information coming from the senses.

25      And, in fact, Antiphon's fragments have been interpreted in very different ways. A rapid summary with an interesting interpretation (which I do not share) may be found in Furley (1981).

26      This was a pressing topic of the age because of the confrontation between democrats and oligarchs in many Greek cities, and many authors wrote on it. See for instance Antiphon (F44a–71); Gorgias (B8a); Thrasymachus (B1, 31); Democritus (B250); see also Protagoras in Plato, *Protagoras* 322c.

27      DK 87 B61. In the *Anonymus Iamblichi*, perhaps a work of Antiphon himself, written in the years of the Peloponnesian war, we find a similar statement: "Respect for the laws [*eunomia*] is the best thing in public and in private, whereas lack of respect for the laws [*anomia*] is the worst, for *anomia* generates tyranny" (DK 89 A 7, 7 and 12).

28      Interestingly, Xenophon (*Hellenica* 2.3.1) reports that the Athenians refer to the year 404 BCE, when the Archon Eponymous was elected under the oligarchy of the Thirty, as the year of *anarchia*.

29      For example, "The human being, who on the one hand claims to be the most godlike of all the beasts" (DK 87 B48).

30      For example, "It is impossible to retract one's life like a move in checkers" (DK 87 B52). See also the sayings attributed to Antiphon in Stobaeus' *Anthology*.

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
