# Peer review of "The Cosmopolitanism of the Early Sophists: The Case of Hippias and Antiphon"

_humanities, doi:10.3390/h12020030_

Round 1

Reviewer 1 Report

The paper is excellent in its current form. I have only a few suggestions.

Line 57: "identitarian"- Does this word make sense in an ancient context?

Lines 132-6: Is the source Plato? If so, the author might note the possibility of irony, as in line 162.

Lines 213-5: This step in the argument does not seem exclusive to sophists, but one made also by philosophers.

Line 348 seems to lightly conflict with line 89: if the title was oft-used, is it still provocative? 

Line 389 makes reference to an idea in Strabo, which is not cited. Other scholars may disagree with this proposed origin of the word for barbarian, so at the very least Strabo should be referenced.

Line 413: Are the perspectives "many" if they are always a pair opposites? A distinction between plural and double seems relevant here.

Author Response

First of all, many thanks to intelligent and very accurate reviewer, I appreciated the painstaking attention to both line of argument and details. here are my answers:

Line 57. I think the word 'identitarian' makes sense in this context. The Greek cities had a strong sense of identity, which sometimes issued in a sense of superiority of one over others.

Lines 132-6. The source is Philostratus and i believe it is not ironic.

Lines 213-5. I agree.

Line 348. Very subtle: point taken, I deleted 'provocatively'.

Line 389. I agree: I added a footnote.

Line 413. Very subtle again: point taken, I changed the text.

Reviewer 2 Report

This is an informative, clearly written and well organized article, which makes the thought of the sophists, in their historical context, accessible to readers not specializing in ancient Greek philosophy and history and is successful in showing the “modernity”, so to speak, of the sophists’ main ideas. 

In particular, the article makes a case for the ascription of a kind of cosmopolitanism to at least two sophists, Hippias and Antiphon, who, in the author’s view, anticipate the explicit cosmopolitanism attributed by ancient tradition to the Cynics. This is not a new thesis, but the author makes an elegant case for it in a way that makes the stakes and potential challenges of this interpretation clear to non-specialized readers. The article draws on a good array of ancient sources as well as the few fragments from the sophists that have been preserved.

I recommend this paper for publication and I suggest some very minor revisions, which are mostly of an editorial kind. In particular:

1.     The article lacks a conclusion. I would recommend the author to draft a general conclusion for the paper and to move there the short conclusion that is now placed at the end of the section devoted to Hippias.

2.     Page 1: a transitional sentence would be necessary between the brief mention of Diogenes of Sinope and the turn to the sophists (perhaps something of the kind: “While the phrase “citizen of the world” does not appear in the available sources concerning the sophists, in this paper I will argue that at least two sophists anticipate Diogenes’ stance, etc. etc.”)

3.     P. 2: it should be Attica rather than Attika

4.     P. 3: This is a more conceptual point: As the author uses the phrase “truth of the matters” in reference to the sophists, perhaps a sentence is here necessary to clarify that the sophists, despite writing treatises On Truth, challenged the correspondence view of truth (as demonstrated by Protagoras’ relativism and Gorgias’ claim that being cannot be communicated), otherwise the reader may be misled or confused.

5.     P. 6: The author should add a note with the sources that refer to the description of Hippias as a traveler 

6.     P. 7: The sentence “Socrates then asks Hippias… and people have to resort to persuasion” is not entirely clear.

7.     In the section on Antiphon, if I correctly understand, the author takes phusis to be a normatively loaded concept. Mauro Bonazzi, on the contrary, in The Sophists argues that nature is value-neutral in Antiphon. Perhaps a footnote acknowledging this difference in interpretation may be useful. Similarly, the question of the sophists' cosmopolitanism is a contested one, hence a footnote (or more) referring to the different positions in this regard so as to better situate the author's thesis within this debate may be in order. 

Author Response

First of all, many thanks to reviewer for the close reading and the excellent, insightful suggestions. I accepted them all. More specifically:

  1. I added a conclusion based on the paragraph suggested by the reviewer.
  2. I added a phrase.
  3. I changed the spelling.
  4. I altered the phrase to clarify my thought.
  5. I added a note specifying the source.
  6. I have changed the sentence in question, I think it is clearer now.
  7. I added the mention to Bonazzi's work and on the debate about the cosmopolitanism of the sophists.

Again, many thanks to the reviewer, it is a pleasure to have such attentive readers.